# Insecticidal Effects of Native Raw and Commercial Diatomaceous Earth Against Lesser Grain Borer and Granary Weevil Under Different Environmental Conditions

**DOI:** 10.3390/insects16060549

**Published:** 2025-05-22

**Authors:** Ayhan Ogreten, Sedat Eren, Cetin Mutlu, Tarkan Ayaz, Amna Saeed, Georgina V. Bingham, William R. Morrison

**Affiliations:** 1Plant Protection Research Institute, Diyarbakir 21110, Türkiye; ayhanogreten@gmail.com (A.O.); erensedat64@hotmail.com (S.E.); 2Department of Plant Protection, Faculty of Agriculture, Harran University, Sanliurfa 63300, Türkiye; amnasaeed067@yahoo.com; 3Department of Entomology, University of Nebraska-Lincoln, Lincoln, NE 68588, USA; gbingham@unl.edu; 4Department of Plant Protection, Faculty of Agriculture, Sirnak University, Sirnak 73000, Türkiye; tarkanayaz@gmail.com; 5Center for Grain and Animal Health Research, Agricultural Research Service, USDA, Manhattan, KS 66502, USA; william.morrison@usda.gov

**Keywords:** *Rhyzopertha dominica*, *Sitophilus granarius*, diatomaceous earth, IPM, storage pests

## Abstract

Wheat is an essential food crop; however, its proper storage is challenging because of insect infestations. The lesser grain borer (*Rhyzopertha dominica*) and the granary weevil (*Sitophilus granarius*) are two significant pests that inflict serious damage to stored wheat. These insects reduce not only the grain quantity, but also quality, leading to economic losses and food waste. Insecticides are commonly used for the management of these insects; however, their usage can also negatively impact human health and the environment. Diatomaceous earth (DE) treatments are safer and eco-friendlier alternatives to insecticides when used correctly. Their effectiveness is directly linked to the temperature, humidity, and application dose. This study evaluated the efficacy of two native DE types from Türkiye and one commercial DE product against *R. dominica* and *S. granarius* under different environmental conditions. The results revealed that the highest test doses of each DE treatment exhibited higher efficacy at high temperatures and a low humidity. Overall, natively sourced Aydın had the highest efficacy. These findings suggest that native Turkish DE treatments could be a useful alternative to conventional insecticides for managing these insects in stored wheat, particularly under conditions similar to those found in Turkish farm storage facilities. This approach offers a natural and practical solution to reduce pest damage without relying on conventional insecticides.

## 1. Introduction

Wheat is a globally significant staple crop, essential to the food security of several countries [1,2]. A significant proportion of the world’s population consumes wheat in the form of bread, pasta, pastries, and cereals as a source of daily calories and protein [3]. However, it is a difficult commodity to store [4] as pest infestation is a serious hazard [5] to the quantity and quality of wheat as well as to other cereal grains stored for later consumption by people and animals [6]. Pest infestation during storage results in substantial economic losses (>USD 100 billion globally) [7], which undermines food security, exacerbates food waste, and exerts adverse impacts on human health [4,8,9]. The feeding of insects on wheat reduces their weight, decreases nutrient contents, and contaminates them with frass, leading to the development of fungi, mycotoxins, and other microorganisms [4,9]. Thus, the consumption of infested grains could lead to significant human health issues [9].

Wheat and stored cereal grains have two main pests, the first is the lesser grain borer [*Rhyzopertha dominica* (F.) (Coleoptera: Bostrychidae)] [10,11], a problem in numerous regions of the world [12,13]. Whole, intact grains are primary host of *R. dominica* [11] and its infestation leads to economic losses for farmers, grain processors, and exporters [4]. Granary weevil [*Sitophilus granarius* L. (Coleoptera: Curculionidae)] is another storage pest infesting and causing significant damage to stored wheat [14,15,16].

Several strategies are used to manage pest infestation in stored wheat [17,18,19]. Integrated pest management (IPM) practices are employed for monitoring, preventing, and managing pest infestation. These measures include the examination and sanitation of storage facilities [20], adherence to appropriate temperature and humidity thresholds [21], and the use of diatomaceous earth (DE) [22,23,24], pesticides [25,26], or fumigation [27,28]. However, the use of pesticides is being re-evaluated due to their adverse impacts on workers, the environment, and non-target organisms [16,29,30]. In this context, the use of DE has been gaining popularity for two decades [22].

DE (also known as diatomite) is a naturally occurring sedimentary rock composed of fossilized remnants of minute algae, termed diatoms [31]. They are commonly used for the management of stored grain pests [16,29,30], owing to their unique physical properties and negligible environmental impacts [22,32], and low mammalian toxicity [22]. DE can provide long-term control over target species as they stay dry and undisturbed after application. The main drawback is that they are usually only effective at higher doses, which can reduce the bulk density of stored grains [33]. However, DE treatments can be safely manipulated without any known resistance in insect pests [34].

Temperature and humidity are the major environmental factors influencing the insecticidal efficacy of DE [22]. They adhere to the cuticle of the insects, causing desiccation and eventually mortality [35]. The relative humidity significantly alters the efficacy of DEs [33,36,37,38]. The efficacy of DEs may be reduced under higher relative humidity since insects can absorb water from the air, and the general efficacy is lessened once the relative humidity reaches 70–75% [22]. In contrast, some studies have also indicated that a higher relative humidity had little impact on the effectiveness, implying that DE treatments exhibit a minimal interaction with moisture [30,39]. Hence, there is no golden rule for the impact of the relative humidity on the efficacy of DE.

In addition, the efficacy of DE treatments significantly fluctuates under varying temperatures [33,38,40,41]. Higher efficacy is observed under elevated temperatures as insects dehydrate rapidly. The improved moisture-absorbing capacities of DE at higher temperatures speed up the desiccation process, leading to high mortality [37,40,42,43]. Therefore, the temperature must be optimized before the application of DE.

The dose is another important factor determining the efficacy of DE. Most DE treatments are effective under higher doses, i.e., 1000–3500 ppm [33]. The application of DE at an appropriate dose is crucial for effective insect control [44]. The pests are then killed when the DE particles absorb through the waxy covering on their exoskeleton, leaving them dehydrated and unable to move [35]. Therefore, it is crucial to apply the optimum dose in order to obtain the desired effects.

Although native and commercial DE treatments have been used to control *R. dominica* [45,46,47,48,49] and *S. granarius* [16,43,50,51], the interactive effect of the temperature, relative humidity, and doses on the efficacy of native raw and commercial DE treatments against lesser grain borer and granary weevil has been less studied. We hypothesized that higher concentrations of DE would exhibit the highest efficacy under a low humidity and high temperature. The results of this study will help to optimize native raw DE treatment doses, temperature, and relative humidity for higher/optimal efficacy.

## 2. Materials and Methods

### 2.1. Experimental Site

This study was conducted at Entomology laboratory, Diyarbakır Plant Protection Research Institute, Diyarbakır Türkiye.

### 2.2. Diatomaceous Earth Treatments

The present study evaluated the efficacy of different doses of three DE treatments, e.g., natively-sourced Aydın and Ankara, and commercially available Silico-Sec on the mortality of lesser grain borer and granary weevil under different temperature and relative humidity conditions. Ankara DE was collected from the mineral deposits located in the Ankara province of Turkey. The composition of the DE was 92.8% SiO_2_, 4.2% Al_2_O_3_, 1.5% Fe_2_O_3_, 0.6% CaO, 0.3% MgO, and a variable amount of water ranging from 1% to 5%. The mean particle size ranged from 8 to 12 μm. The Aydın DE was obtained from the mines located in the Aydın province of Turkey. The composition of the DE was 94.2% SiO_2_, 4.6% Al_2_O_3_, 1.6% Fe_2_O_3_, 0.7% CaO, 0.3% MgO, and a variable amount of water ranging from 1–5%. The mean particle size ranged from 8 to 12 μm. Silico-Sec (a product of Biofarma, GmbH, Lübbecke, Germany). is currently available commercially in Germany. It is composed of 92% SiO_2_, 3% Al_2_O_3_, 1% Fe_2_O_3_, and 1% Na_2_O. The mean particle size ranges from 8 to 12 μm. The native raw DEs have comparable properties to the commercially available DE. Therefore, we were interested to know whether the native raw DE treatments can cause similar mortality in lesser grain borer and granary weevil compared to the commercially available DE.

### 2.3. Study Insects

The test insects included 1000 adults for each species with no prior exposure to insecticides, and they were collected from the laboratory population at Plant Protection Research Institute Diyarbakir, Türkiye. The insects were cultured on wheat grains in 1000 mL plastic containers under 25 ± 1 °C, 65 ± 5% relative humidity, and complete darkness in environmental chambers. The cultures were maintained under identical conditions to produce subsequent generation adults. The newly emerged adults, of the same age, were collected using a suction tube and used in the bioassays below.

### 2.4. Bioassays

Transparent plastic containers (200 g capacity) were used in the bioassays. Five doses, i.e., 0, 250, 500, 750, and 1000 ppm, of all DE treatments were included in this study. The whole grains of the ‘Pehlivan’ bread wheat variety were sterilized by exposing grain to 55 °C for 48 h. Subsequently, 100 g of sterilized grains were placed in plastic containers with weighed amounts of each DE according to the doses below that were added to grains. Jars were subsequently shaken for 2 min to thoroughly mix the DEs. The control grains did not receive any DE. A total of 30 adults of each species (separately) were then released in each jar. Each treatment had 10 replications, and the experiment was repeated over time.

The jars were kept in incubators maintained at 25 °C or 30 °C with 40% or 60% relative humidity at each temperature. Dead and alive adults were counted at 7, 14, and 21 d after their release. Temperature and relative humidity were monitored with a Testo 174H brand temperature/humidity data logger (Titisee-Neustadt, Germany). The mortality data of the treatments were adjusted by Abbott’s formula and used in the statistical analysis.

### 2.5. F_1_ Progeny Production

The live adults (if any) were collected after 21 d of their release and kept under the same conditions as above for ~2 months to determine F_1_ progeny production.

### 2.6. Statistical Analysis

The data of both insect species were analyzed and presented separately. An analysis of variance (ANOVA) was used to analyze the mortality data [52]. The normality and homogeneity of variance was evaluated before analyzing the data with ANOVA. The data were normally distributed; therefore, all analyses were performed on untransformed data. The differences between the experiments were inferred with post hoc *t*-tests, which were not significant. Therefore, data for both experiments were pooled for the final analysis. Three-way ANOVA was used to test the significance of main and interactive effects of temperature, relative humidity, and doses of DE treatments on mortality and progeny production. The least significant difference at 95% probability was used as a post hoc test to separate the means upon a significant result from the ANOVA. Statistical analyses were conducted using SPSS statistical software version 21 [53]. All individual and interactive effects were significant; therefore, only three-way interactions were presented and interpreted in this study.

## 3. Results

The individual and interactive effects of the temperature, relative humidity, and doses of all DE treatments significantly affected the mortality of *R. dominica* at 7, 14, and 21 DAT (Table 1).

The mortality linearly increased with time after treatment. Overall, the 1000 ppm dose of Aydın DE caused the highest mortality in *R. dominica* at all sampling dates, which was followed by Silico-Sec and Ankara DE, respectively (Figure 1).

The three-way interaction of the temperature, relative humidity, and DE doses indicated that a 1000 ppm dose of Aydın DE under a 30 °C temperature and 40% relative humidity caused the highest mortality in *R. dominica* at all data collection dates (Figure 2). The 1000 ppm dose of Ankara DE caused 31% mortality in *R. dominica* at 7 DAT under 30 °C and 40% relative humidity, which increased to 52% and 69% at 14 and 21 DAT, respectively. Similarly, the 1000 ppm dose of Aydin DE caused 38% mortality in *R. dominica* at 7 DAT under the same conditions, which increased to 64% and 85% at 14 and 21 DAT, respectively. Likewise, the 1000 ppm dose of Silico-Sec resulted in the 37% mortality of *R. dominica* at 7 DAT under 30 °C and 40% relative humidity, which increased to 61% and 81% at 14 and 21 DAT, respectively (Figure 2). The lowest dose (0 ppm) of all DE treatments caused the least mortality under a 25 °C temperature and 60% relative humidity at all data collection intervals (Figure 2).

The mortality of *S. granarius* was significantly influenced by the individual and interactive effects of the temperature, relative humidity, and DE doses at 7, 14, and 21 DAT (Table 2).

The mortality also linearly increased with time for *S. granaries,* as observed in *R. dominica*. Overall, the 1000 ppm dose of Aydın DE caused the highest mortality in *S. granarius* at all sampling dates, which was followed by Silico-Sec and Ankara DE, respectively (Figure 3).

The three-way interaction between the temperature, relative humidity, and DE doses denoted that the highest mortality in *S. granarius* was recorded with the 1000 ppm dose of Aydın DE under a 30 °C temperature and 40% relative humidity (Figure 4). The 1000 ppm dose of Ankara DE under a 30 °C temperature and 40% relative humidity caused 70% mortality in *S. granarius* at 7 DAT, which increased to 74% and 100% at 14 and 21 DAT, respectively. Similarly, the 1000 ppm dose of Aydin DE under 30 °C temperature and 40% relative humidity resulted in 86% mortality in *S. granarius* at 7 DAT, which increased to 91% and 100% after 14 and 21 DAT, respectively. Likewise, the 1000 ppm dose of Silico-Sec under the same conditions caused 83% mortality in *S. granarius* 7 DAT, which increased to 88% and 100% after 14 and 21 DAT, respectively (Figure 4). The lowest mortality was observed for 250 ppm doses of all DE treatments under a 25 °C temperature and 60% relative humidity at all data collection intervals (Figure 4).

The live insects were observed for 2 months to record F_1_ progeny. However, no adult emergence was recorded in any of the treatments included in this study.

## 4. Discussion

The results showed that tested DE treatments caused the significant mortality of both species under varying climatic conditions included in this study. Overall, each of the DE treatments caused higher mortality in *S. granarius* compared to *R. dominica*. The mortality of *S. granarius* was >70% at 7 DAT, indicating that the DE could be included as a tool within integrated pest management programs. On the other hand, the mortality of *R. dominica* was <40% at 7 DAT for all of the DE treatments, suggesting the lower susceptibility of this species. The mortality of *R. dominica* reached 60% at 14 DAT, while it was >85% in *S. granarius*. The differences among species may be attributed to their body size, morphological structure, and movement patterns. Body size has been reported to significantly influence the efficacy of insecticides [54]. *Sitophilus granarius* exhibits a larger body size (ranging from 3–5 mm) [55] compared to *R. dominica* (ranging from 2.5–3 mm) [56] and possesses distinct morphological characteristics such as numerous dentitions on the pronotum and elytra, deep pits, and short, dense hairs. The larger body size probably provided a larger surface area for DE contact, which resulted in the higher mortality of *S. granarius* compared to *R. dominica*. In addition, *Sitophilus* spp. are often more mobile than *R. dominica*, at least in terms of walking patterns [57], so likely more quickly accumulate DE on their cuticle.

Different temperatures, relative humidity levels, and DE doses significantly altered the mortality of both species as hypothesized. All DE treatments caused higher mortality at their highest doses in this study under 30 °C and 40% RH. Several studies have reported that the efficacy of DE is dependent on the target species and environmental conditions after the application of DE [37,40,41,58,59,60,61]. Moreover, the nature and properties of DE treatments are another strong driver affecting their efficacy [32,60,62]. The major aim of this study was to infer whether the natively sourced raw DEs could cause comparable mortality to commercial DE. The results revealed that the native raw Aydin DE caused similar and, in some cases, higher mortality in both species than the commercial Silico-Sec DE in the current study. Therefore, the native DE Aydin could be successfully used to control both stored pests of wheat grains in the country. Several earlier studies have indicated comparable efficacies of native raw DE with commercially available ones [38,40,41].

The mortality of both species significantly increased by increasing the DE doses and exposure duration. The higher doses (1000 ppm) of all DE treatments caused greater mortality to both species under a high temperature (30 °C) and low relative humidity (40%). Rapid dehydration in adults by absorbing lipids and disrupting cuticular waxes at a low humidity and high temperature is thought to be the reason for high mortality. Several earlier studies have reported the enhanced effectiveness of DE at higher temperatures [37,38,40,43,50]. Increasing the mobility of insects in response to rising temperatures and higher water loss are the main reasons for the increased mortality of the stored pest species in the current study. Moreover, elevated temperatures are expected to lead to increased water loss [33,37,43,45]. Although most of the studies have indicated that higher temperature provides better efficacy with DE, a recent study indicated that sometimes higher efficacy could be observed under a low temperature [63]. Stress caused by the low temperature and further stress by the application of DE is reported to be the reason for higher mortality under low-temperature conditions [63].

The insecticidal efficacy of the tested DE treatments significantly varied despite the same range of particle sizes. The Aydin DE caused the highest mortality followed by Silico-Sec and Ankara. The results indicated that the efficacy of DE treatment is dependent upon their source, likely as a function of their composition and enrichment with silica dioxide. Higher SiO_2_ and Al_2_O_3_ concentrations and slightly higher Fe_2_O_3_ in Aydın DE may increase its toxicity. These constituents increased oxidative stress and increased physical damage, both of which could contribute to higher mortality in the studied species. Scanning electron microscopy (SEM) imaging and zeta potential provide reliable information relating to the morphology and surface structure and surface charge of the particles, respectively. However, these were not studied. Future studies should consider the use of SEM imaging and zeta potential to further understand the increased insecticidal activity of Aydın DE. Golob [64] reported differences in the efficacies of DE across geological regions. Furthermore, studies have documented variations in the insecticidal properties of DE derived from diverse geographic locations and mining sites [38,40]. Korunic [60] conducted a study on the insecticidal effectiveness and bulk density of 25 distinct DE types in stored commodities. The findings revealed that the insecticidal activity and bulk density of the DE varied significantly. The literature suggests that the diverse effects of numerous DE formulations are likely attributed to variations in their morphological and physical properties [33,65].

The current study indicates that the application of 1000 ppm for all DE treatments provided the highest mortality rate by both species. The 1000 ppm is equivalent to 1 kg DE per ton of grain, and this is the recommended dose for commercial DE [32,39,66,67]. Therefore, from the results shown here, native raw DE Aydin is a suitable candidate for use in the management of both stored pest species.

The effectiveness of DE is heavily influenced by the relative humidity owing to their significant absorptive capacity [22]. However, some studies have reported that the efficacy of DE remains unaffected by the relative humidity [43]. Some types of DE may exhibit reduced efficacy under high humidity, compared to their performance under dry conditions. A decrease in the efficacy of DE has been reported with an increasing relative humidity from 55% to 65% against *Tribolium confusum* Jacquelin du Val (Coleoptera: Tenebrionidae) [22]. Appropriate relative humidity levels are crucial for protecting grains during storage. Relative humidity levels from 55% to 75% correspond to 10.5% to 14% moisture in the stored grain under an equilibrium, which is practical and feasible for extended storage periods [22]. Ogreten et al. [68] have recently reported that all DE treatments included in the current study caused higher mortalities in *T. confusum* under a 30 °C temperature and 40% relative humidity. The current study also revealed that the tested DE treatments were more effective at a lower relative humidity. Hence, the optimum relative humidity level for the higher efficacy of the tested DE in the current study would not have a negative impact on the stored grains. Storage facilities in Türkiye have similar temperature and relative humidity levels during wheat storage [69]. Therefore, the DE treatments could be successfully used to control both tested species in stored wheat.

## 5. Conclusions

The present study indicated that the 1000 ppm dose of all DEs caused higher mortality under a 30 °C temperature and 40% relative humidity. Thus, it is key to utilize the DE treatments in the specified environmental conditions for optimal efficacy. The native raw Aydin DE and the commercial DE Silico-Sec caused equivalent mortality in both species. Therefore, native Aydin DE has the potential to effectively manage both species, *Rhyzopertha dominica* and *Sitophilus granaries,* in stored wheat at the specified doses and environmental conditions.

## Figures and Tables

**Figure 1 insects-16-00549-f001:**
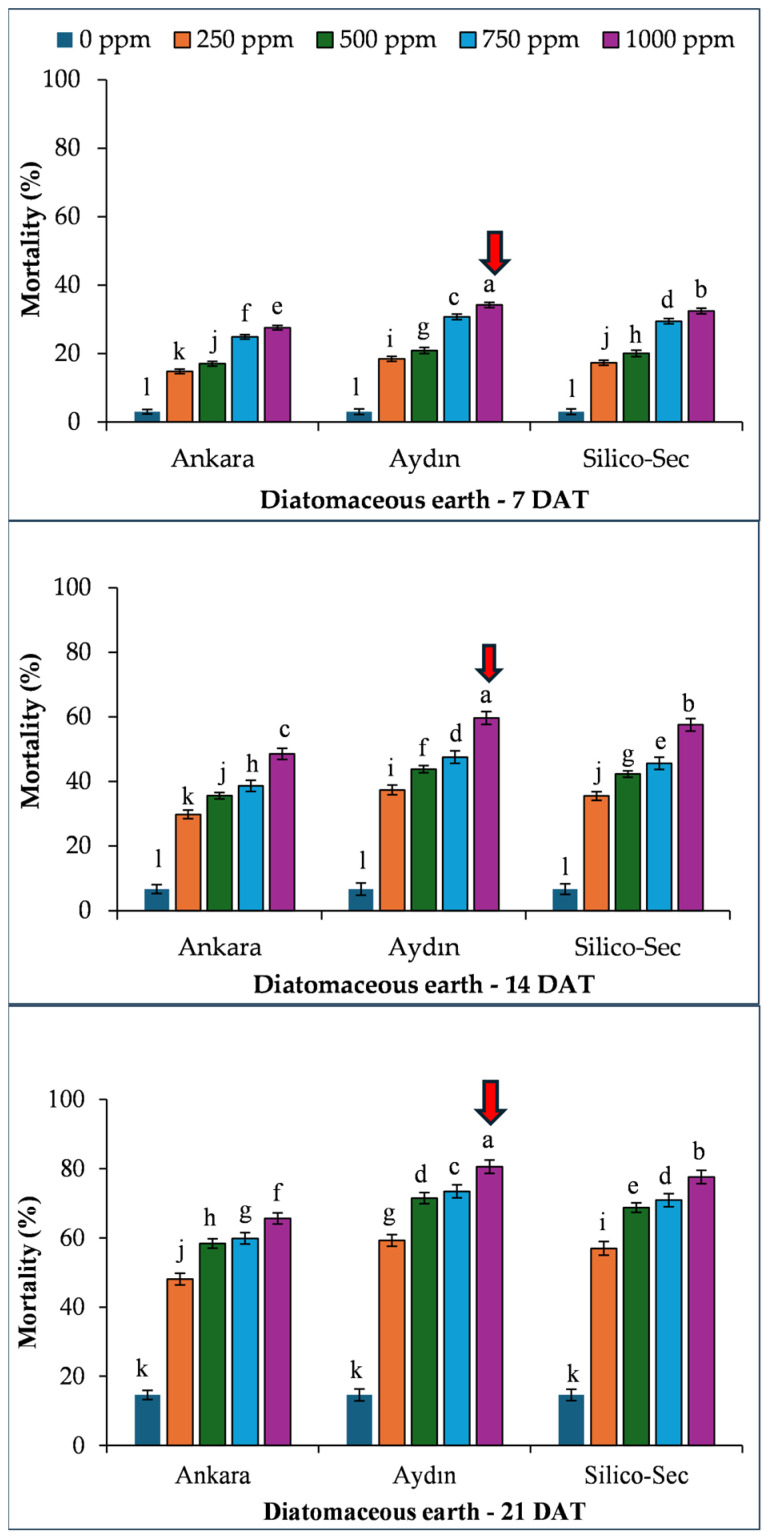
The influence of various diatomaceous earth doses on the mortality of *Rhyzopertha dominica*. The values presented are means ± SEM (n = 10). Bars with shared letters are not significantly different from each other (LSD, α = 0.05). The red arrows indicate the treatment causing the highest mortality of the species.

**Figure 2 insects-16-00549-f002:**
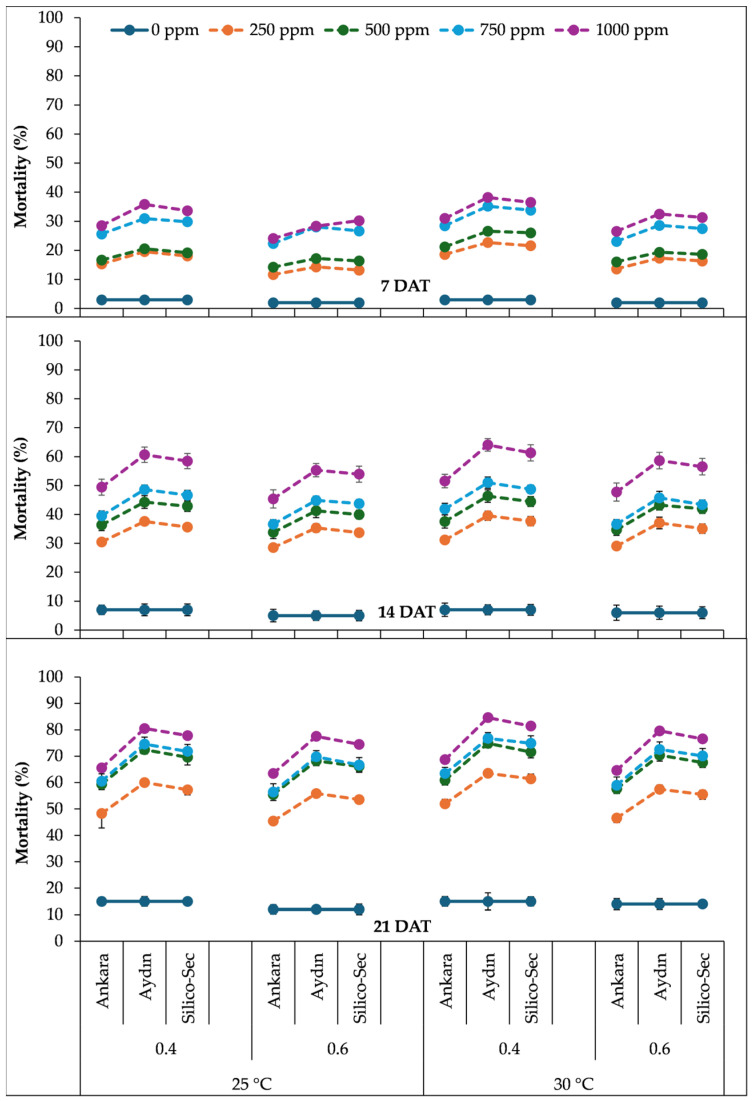
The influence of temperature regimes, relative humidity levels, and doses of DE treatments on the mortality of *Rhyzopertha dominica*. The values presented are means ± SEM (n = 10).

**Figure 3 insects-16-00549-f003:**
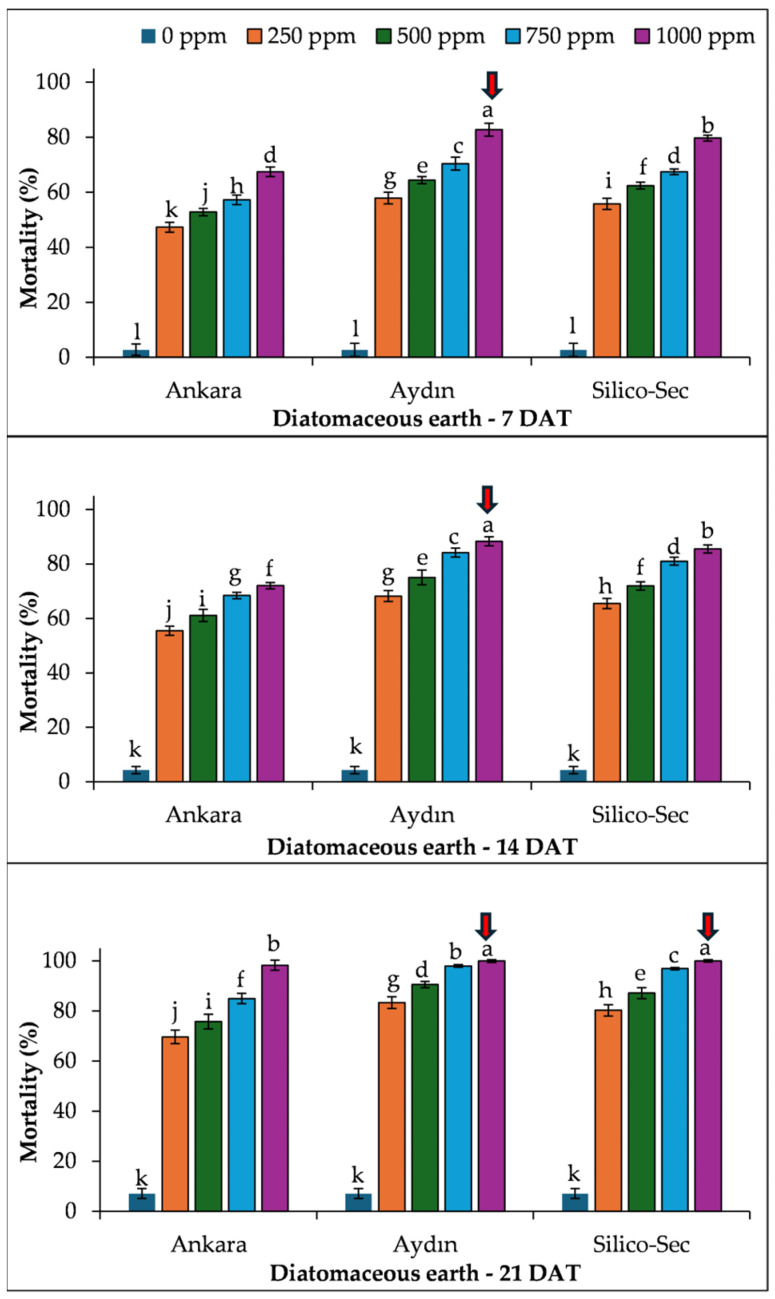
The influence of DE doses on the mortality of *Sitophilus granarius.* The values presented are means ± SEM (n = 10). Bars with shared letters are not significantly different from each other (LSD, α = 0.05). The red arrows indicate the treatment causing the highest mortality of the species.

**Figure 4 insects-16-00549-f004:**
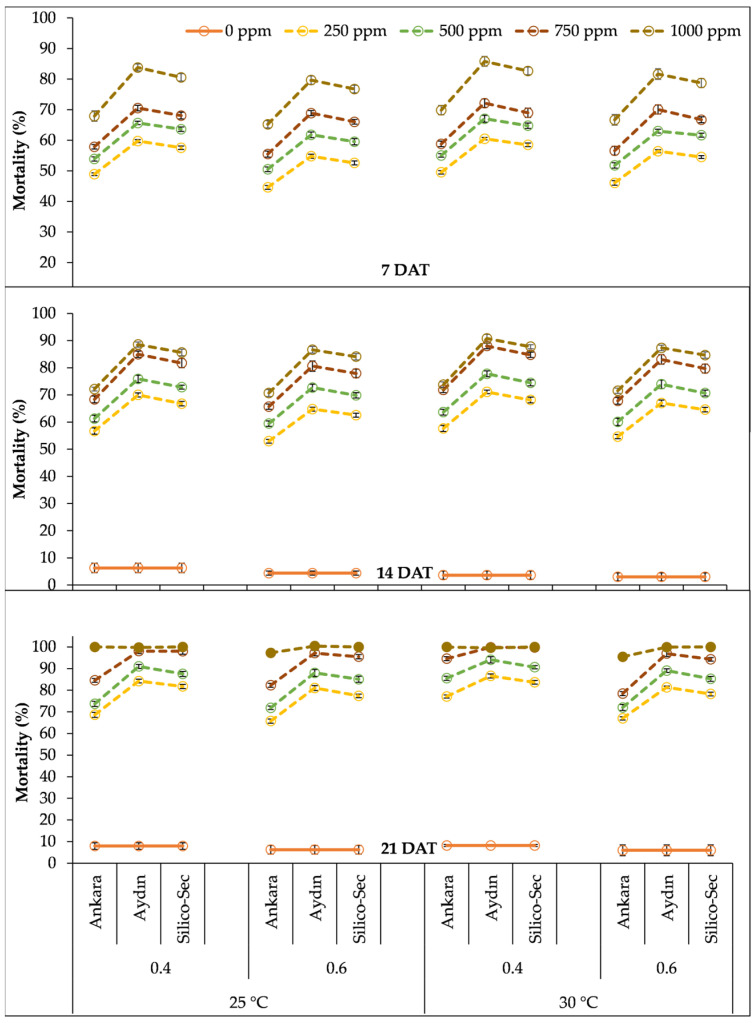
The influence of temperature regimes, relative humidity levels, and doses of DE treatments on the mortality of *Sitophilus granarius.* The values presented are means ± SEM (n = 10).

**Table 1 insects-16-00549-t001:** Analysis of variance (*p* and F values) for the main and interactive effects of different temperature regimes, relative humidity levels, and various doses of diatomaceous earth treatments on the mortality of *Rhyzopertha dominica*.

		7 DAT		14 DAT		21 DAT	
Source of Variation	DF	F Value	*p* Value	F Value	*p* Value	F Value	*p* Value
Temperature (T)	1	1162.35	**<0.0001**	311.91	**<0.0001**	790.45	**<0.0001**
Relative humidity (RH)	1	3266.15	**<0.0001**	1249.44	**<0.0001**	2184.86	**<0.0001**
Diatom dose (D)	12	3242.77	**<0.0001**	5244.91	**<0.0001**	10,006.24	**<0.0001**
T × RH	1	114.15	**<0.0001**	10.54	**0.001**	20.65	**<0.0001**
T × D	12	13.62	**<0.0001**	5.49	**<0.0001**	4.03	**0.005**
RH × D	12	13.84	**0.000**	10.95	**<0.0001**	8.87	**<0.0001**
T × RH × D	12	9.05	**<0.0001**	1.79	**0.030**	5.04	**<0.0001**

The bold values in the *p* value column indicate significant effect of the relative treatment.

**Table 2 insects-16-00549-t002:** Analysis of variance (*p* and F values) for the individual and interactive effects of different temperature regimes, relative humidity levels, and various diatomaceous earth doses on the mortality of *Sitophilus granaries*.

		7 DAT		14 DAT		21 DAT	
Source of Variation	DF	F Value	*p* Value	F Value	*p* Value	F Value	*p* Value
Temperature (T)	1	279.48	**<0.0001**	456.70	**<0.0001**	439.87	**<0.0001**
Relative humidity (RH)	1	1946.92	**<0.0001**	2120.71	**<0.0001**	2878.05	**<0.0001**
Diatom dose (D)	12	17,997.23	**<0.0001**	23,139.83	**<0.0001**	27,038.20	**<0.0001**
T × RH	1	2.73	**<0.0001**	13.64	**<0.0001**	633.84	**<0.0001**
T × D	12	7.02	**<0.0001**	17.57	**<0.0001**	57.76	**<0.0001**
RH × D	12	13.91	**<0.0001**	15.89	**<0.0001**	102.50	**<0.0001**
T × RH × D	12	1.11	**<0.0001**	4.23	**<0.0001**	66.87	**<0.0001**

The bold values in the *p* value column indicate significant effect of the relative treatment.

## Data Availability

The original contributions presented in this study are included in the article. Further inquiries can be directed to the corresponding author.

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
