# Peer review of "Insecticidal Effects of Native Raw and Commercial Diatomaceous Earth Against Lesser Grain Borer and Granary Weevil Under Different Environmental Conditions"

_insects, 2025, doi:10.3390/insects16060549_

Round 1

Reviewer 1 Report

Comments and Suggestions for Authors

Dear Authors, the work is interesting and valuable. I would like to see it again before publishing, so I am forced to give it a "major", but please do not worry. I do not have many comments.

  1. Line 70 „Morrison et al., 2019a” incorrect citation style.
  2. In some places like line 147, 196, 198, 203 etc there is incorrect symbol of degree. There is  “º” (letter o) insetead of the degree symbol “°” .
  3. In the methods there is mentioned control variant (0 ppm) but it is not included on the diagrams (Figure 1, 2, etc).

The study was designed correctly, but the results lack presentation and comparison of the control variant (0 ppm) with other DE doses. Please complete this information (also in the diagrams). Without that we do not know the natural mortality of insects and it is impossible to conclude.

Best regards

Author Response

Response to Reviewer 1

Dear Authors, the work is interesting and valuable. I would like to see it again before publishing, so I am forced to give it a "major", but please do not worry. I do not have many comments.

Response: Thank you for your detailed and constructive criticism which helped us to improve the quality of the manuscript. We have revised the manuscript according to your suggestions and hope that it will be recommended for publication.

  1. Line 70 „Morrison et al., 2019a” incorrect citation style.

Response: Citation style has been corrected.

  1. In some places like line 147, 196, 198, 203 etc there is incorrect symbol of degree. There is  “º” (letter o) insetead of the degree symbol “°” .

Response: Degree signs have been corrected throughout the manuscript.

  1. In the methods there is mentioned control variant (0 ppm) but it is not included on the diagrams (Figure 1, 2, etc).

Response: We agree with you that there is 0 ppm dose in the methods. This dose has been added in the figures.

The study was designed correctly, but the results lack presentation and comparison of the control variant (0 ppm) with other DE doses. Please complete this information (also in the diagrams). Without that we do not know the natural mortality of insects and it is impossible to conclude.
Response: We agree with you that there is 0 ppm dose in the methods. This dose has been added in the figures.

Best regards

Reviewer 2 Report

Comments and Suggestions for Authors

The study was aimed at Insecticidal effects of native raw and commercial diatomaceous earths against lesser grain borer and granary weevil under different environmental conditions. The pests of focus are R. dominica and S. granaries, both of which causes significant post-harvest losses in wheat. Given the drawbacks if synthetic insecticides, the present study investigates native Turkish DEs as viable eco-friendly alternatives to the commercial product Silico-Sec

The research question is interesting and shows relevance to the field of stored product pest management. While DEs have been widely studied, comparative efficacy of native Turkish DEs under varying environmental conditions against key pests had not been extensively explored. Thus, the present study adds local specificity and expands knowledge on the interaction of DE origin, dose and abiotic factors like temperature and humidity. However, the manuscript can be accepted for publication with minor revisions.

  1. Is the grain quality assessed post-treatment?
  2. What could be the reason for Aydın DE to cause higher mortality in both species? I would suggest including the particle characterization studies like SEM imaging, zeta potential for both Ankara and Aydın DEs.
  3. The manuscript needs extensive English language editing for professional presentation. For Instance,

Line 44- Change “Survivors of both species….” To “No F1 progeny were produced by surviving individuals of both species”

Line 63- change to ‘the primary host’

Line 70-reference format inconsistent, use numeric style for uniformity.

Line 85- change to ‘DEs adhere’

Line156- change to ‘Abbott’s

Line 253- change ‘different DEs’ to ‘tested DEs’

Line 257-change “indicating that the species is less sensitive to the tested DEs’ to ‘suggesting lower susceptibility of this species to the tested DEs’

Figure 2- maintain uniform scale on y-axis

Reference- please provide doi for all the references to maintain uniformity.

Author Response

Response to Reviewer 2

The study was aimed at Insecticidal effects of native raw and commercial diatomaceous earths against lesser grain borer and granary weevil under different environmental conditions. The pests of focus are R. dominica and S. granaries, both of which causes significant post-harvest losses in wheat. Given the drawbacks if synthetic insecticides, the present study investigates native Turkish DEs as viable eco-friendly alternatives to the commercial product Silico-Sec.   

Response: Thank you for your detailed and constructive criticism which helped us to improve the quality of the manuscript. We have revised the manuscript according to your suggestions and hope that it will be recommended for publication.

The research question is interesting and shows relevance to the field of stored product pest management. While DEs have been widely studied, comparative efficacy of native Turkish DEs under varying environmental conditions against key pests had not been extensively explored. Thus, the present study adds local specificity and expands knowledge on the interaction of DE origin, dose and abiotic factors like temperature and humidity. However, the manuscript can be accepted for publication with minor revisions.

Response: Thank you for encouraging comments. The manuscript has been thoroughly edited for language and grammar.

Is the grain quality assessed post-treatment?

Response: No, the grain quality was not tested unfortunately.

What could be the reason for Aydın DE to cause higher mortality in both species? I would suggest including the particle characterization studies like SEM imaging, zeta potential for both Ankara and Aydın DEs.

Response: Higher SiO2 and Al2O3 concentrations in Aydın DE, together with its slightly higher Fe2O3 values, might increase its toxicity. The chemical constituents may result in increased oxidative stress (attributable to iron oxide) and increased physical damage (due to silica and aluminium oxides), both of which might contribute to higher mortality in the studied species. We don’t have SEM imaging and zeta potential data. We have suggested that future studies should explore these to have a better understanding of the increased insecticidal activity of Aydın DE.

The manuscript needs extensive English language editing for professional presentation. For Instance,

Line 44- Change “Survivors of both species….” To “No F1 progeny were produced by surviving individuals of both species”

Response: Changed as suggested

Line 63- change to ‘the primary host’

Response: Changed as suggested

Line 70-reference format inconsistent, use numeric style for uniformity.

Response: Citation corrected

Line 85- change to ‘DEs adhere’

Response: Changed as suggested

Line156- change to ‘Abbott’s

Response: Changed as suggested

Line 253- change ‘different DEs’ to ‘tested DEs’

Response: Changed as suggested

Line 257-change “indicating that the species is less sensitive to the tested DEs’ to ‘suggesting lower susceptibility of this species to the tested DEs’

Response: Changed as suggested

Figure 2- maintain uniform scale on y-axis

Response: The Y-axis scale has been uniformed as suggested

Reference- please provide doi for all the references to maintain uniformity.

Response: DOIs have been added for the papers where available.

Round 2

Reviewer 1 Report

Comments and Suggestions for Authors Dear Authors,
All my comments have been taken into account. I see that the manuscript has been improved in many other places. I think that this form is appropriate and therefore can be published.
Best regards